# Deep Learning Applied to Raman Spectroscopy for the Detection of Microsatellite Instability/MMR Deficient Colorectal Cancer

**DOI:** 10.3390/cancers15061720

**Published:** 2023-03-11

**Authors:** Nathan Blake, Riana Gaifulina, Lewis D. Griffin, Ian M. Bell, Manuel Rodriguez-Justo, Geraint M. H. Thomas

**Affiliations:** 1Department of Cell and Developmental Biology, University College London, London WC1E 6BT, UK; 2Department of Computer Science, University College London, London WC1E 6BT, UK; 3Spectroscopy Products Division, Renishaw PLC, Wotton-under-Edge GL12 8JR, UK; 4Department of Research Pathology, Cancer Institute, University College London, London WC1E 6DD, UK

**Keywords:** Raman spectroscopy, deep learning, oncology, microsatellite instability, diagnostics, colorectal cancer

## Abstract

**Simple Summary:**

Colorectal cancer has several disease pathways which have implications for how patients are monitored and treated. One important pathway is caused by deficiencies to genes responsible for repairing pre-cancerous cells. Methods to detect these deficiencies exist, but are not implemented as often as recommended and could be improved. Raman spectroscopy is a technique that could provide such an improvement, having shown potential in other areas of cancer research. The full potential of Raman datasets may be achieved by exploiting modern machine learning models. We evaluated a small colorectal tissue dataset to assess the viability of some common machine learning techniques to detect different colorectal cancer pathways. We find that Raman spectroscopy in conjunction with machine learning could be a viable means of improving screening and potentially diagnostic tools and warrants further research with larger sample sizes.

**Abstract:**

Defective DNA mismatch repair is one pathogenic pathway to colorectal cancer. It is characterised by microsatellite instability which provides a molecular biomarker for its detection. Clinical guidelines for universal testing of this biomarker are not met due to resource limitations; thus, there is interest in developing novel methods for its detection. Raman spectroscopy (RS) is an analytical tool able to interrogate the molecular vibrations of a sample to provide a unique biochemical fingerprint. The resulting datasets are complex and high-dimensional, making them an ideal candidate for deep learning, though this may be limited by small sample sizes. This study investigates the potential of using RS to distinguish between normal, microsatellite stable (MSS) and microsatellite unstable (MSI-H) adenocarcinoma in human colorectal samples and whether deep learning provides any benefit to this end over traditional machine learning models. A 1D convolutional neural network (CNN) was developed to discriminate between healthy, MSI-H and MSS in human tissue and compared to a principal component analysis–linear discriminant analysis (PCA–LDA) and a support vector machine (SVM) model. A nested cross-validation strategy was used to train 30 samples, 10 from each group, with a total of 1490 Raman spectra. The CNN achieved a sensitivity and specificity of 83% and 45% compared to PCA–LDA, which achieved a sensitivity and specificity of 82% and 51%, respectively. These are competitive with existing guidelines, despite the low sample size, speaking to the molecular discriminative power of RS combined with deep learning. A number of biochemical antecedents responsible for this discrimination are also explored, with Raman peaks associated with nucleic acids and collagen being implicated.

## 1. Introduction

Colorectal cancer (CRC) encompasses cancers of the large colon and rectum. It is one of the major malignancies of the world, being the third most commonly occurring and the second most deadly cancer, with an estimated 1.8 million new cases and 881,000 deaths worldwide in 2018 [1]. With a few exceptions, CRC incidence is increasing globally, particularly in developing countries, as shifting dietary and lifestyle factors are likely driving an increase in early onset CRC [2].

There are several known pathological pathways leading to CRC, resulting in heterogeneous presentations, therapies and outcomes. One such pathway is DNA mismatch repair deficient (dMMR) CRC, in which there are pathological alterations to any of a number of MMR genes (MLH1, MSH2, MSH6 or PMS2). This loss of MMR function causes high-level microsatellite instability (MSI-H), characterised by mononucleotide, dinucleotide and trinucleotide tandem repeats. This can occur sporadically or as an inherited trait, as in Lynch syndrome (LS). Hence, the detection of MSI-H is recommended in every case of CRC to screen for LS [3]. The high mutational burden seen in MSI-H tumours also has implications for treatment, providing potential targets for immunotherapy such as immune checkpoint inhibitors [4].

Despite recommendations for universal testing for MSI-H for all CRC cases, resource limitations mean that this cannot always happen and is particularly poor for young adults [5]. Testing for dMMR/MSI-H typically involves either immunohistochemistry (IHC) of the mismatch repair proteins or PCR amplification of consensus microsatellite repeats. Recent developments in machine learning (ML) have led to the possibility of exploiting morphological information in standard H&E slides [4,6]. Such digital pathology techniques require few additional resources and have proven highly accurate in high quality, curated datasets. However, consistent with other domains using modern ML, these promising results do not generalise well when applied to settings or cohorts outside of the context in which they were developed [6]. The literature has thus far focused on H&E stained slides but ML applied to other histological stains, such as IHC, may yield further improvements by exploiting molecular-level information. Raman spectroscopy is a technique which opens the possibility of digitally staining a sample for many biomolecules, without the need for specific sample preparation [7].

Raman spectroscopy (RS) is a technique which interrogates molecular vibrational states through inelastically scattered photons from a sample. As the vibrational modes of any given molecule are unique, it is possible to identify a molecule through its Raman spectrum. A Raman spectrum represents the change in photon wavenumber from a monochromatic light source along the x-axis and the intensity (i.e., number of photons) thus scattered on the y-axis. RS has been successfully applied to discriminate between numerous cancer types in human tissues, most recently to the brain, breast, cervical, colon, lung, nasopharyngeal, prostate, skin and tongue [8]. The applications include early diagnosis, biopsy guidance and intraoperative tumour margin detection. As a molecularly sensitive modality, RS may be able to detect the molecular antecedents of microsatellite stable (MSS) and MSI-H samples.

Despite promising results, RS has yet to become routinely used in the oncology setting for several reasons, including technical limitations in in vivo applications, a lack of visibility in the medical literature and a lack of thorough validation of the models on truly independent datasets [9]. Additionally, the inherent complexity and high dimensionality (number of discrete wavenumber points per spectrum) of biomedical RS data, compounded with various sources of noise, such as fluorescence, necessitates the use of mathematical modelling to extract coherent molecular information.

Deep learning (DL) is a family of modelling techniques under the umbrella of ML models. Its ability to capture non-linear relationships make it suited to complex clinical datasets, and it may help unleash the potential of RS. In addition to being increasingly applied to oncology problems in the context of digital histopathology [10], it is now extending into oncological applications of RS [8]. However, DL is notoriously data-intensive, which makes it difficult to apply to smaller medical proof-of-concept studies. Combined with the large size of DL models, it is susceptible to over-fitting, in which excellent results on a dataset fail to transfer across to general settings. However, with developing techniques such as data augmentation and strict validation practices, it may still be possible to leverage DL even with small datasets.

This proof-of-concept study explores the potential of RS to distinguish between healthy, MSS adenocarcinoma (AC) and MSI-H AC in human tissue. A DL model is developed alongside two traditional ML models commonly used in RS, taking great methodological care not to produce overly optimistic performance estimates. Biochemical antecedents responsible for the DL model’s discriminative performance are then inferred. The exploratory nature of this study seeks only to assess whether RS applied to this particular clinical problem merits additional studies, to highlight some of the methodological considerations for such a larger study and to explore potential clinical uses such as screening or diagnosis.

## 2. Materials and Methods

### 2.1. Tissue Acquisition and Processing

Formalin fixed paraffin embedded (FFPE) human colon samples were obtained from the UCL/UCLH Biobank for Studying Health and Disease (REC 20/YH/0088). A total of 10 FFPE samples of resection margins of normal colonic mucosa from sporadic CRC cases were obtained along with 10 MSS/MMR proficient samples from the same patients. A total of 10 archival MSI-H samples were also obtained and matched to the sporadic AC samples by cancer stage (T-stage, see Appendix D for details).

From each sample, one section was cut at 8 μm thickness and mounted onto silanised 304L super mirror stainless steel slides for Raman analysis and one 3 μm section was cut and mounted onto standard glass slides for H&E staining. The steel mounted samples were prepared as described by Gaifulina et al. [11]. Steel slides have been shown to improve Raman signal acquisition by up to a factor of four and reduce background signal compared to calcium fluoride ($CaF_{2}$), the standard substrates often used in RS [11,12], and are far cheaper.

Unstained tissue sections were immersed in a series of baths to remove paraffin wax. Four successive ten-minute baths in xylene (VWR International Ltd., Lutterworth, UK) with gentle agitation, were followed by a series of rehydration steps in graded ethanol absolute (VWR International Ltd., Lutterworth, UK), followed by a final immersion in distilled water for ten minutes.

The 3 μm sections were subject to standard automated staining and cover-slipping for H&E slides. The histopathology of all samples was confirmed by a resident consultant pathologist. A full breakdown of the patient samples and cancer stages can be found in Appendix D Table A2.

### 2.2. Raman Spectroscopy

Point spectra were acquired using the Renishaw prototype RA800 series benchtop system with a 785 nm laser (Renishaw plc, Wotton-under-Edge, UK). A total laser intensity of approximately 158 mW was focused onto samples through a 50 × NA 0.8 objective. A 1500 L/mm grating was used to disperse the light providing a spectral range of 0–2100 cm${}^{-1}$ in the low wavenumber range. An integration time of 20 s was used for all measurements. A total of 50 individual spectra were collected from each tissue sample, except for one sample with only 40 spectra, resulting in a total of 1490 across the 3 classes. All spectra were acquired from the glandular mucosal region in normal samples and from confirmed cancerous regions in all cancer samples, located by the resident pathologist prior to Raman measurement.

### 2.3. Modelling and Cross-Validation Strategy

Cosmic rays were removed using a combination of the width of feature and nearest neighbour methods available in the manufacturer’s software. Spectra were visually inspected at the time of acquisition and any saturated spectra were discarded and a new spectrum obtained from a different region. Each spectrum was standard normal variate (SNV) normalised to have zero mean and unit variance.

Baseline correction was not performed. An initial analysis showed that baseline correction via several methods (references [13,14,15,16]) did not improve performance, and significantly altered resulting mean and difference spectra which impacted the biochemical interpretation (details in Appendix A).

A principal component analysis–linear discriminant analysis (PCA–LDA), a support vector machine (SVM) and a convolutional neural network (CNN) were developed using SNV normalised Raman spectra truncated to the range of 400–1800 cm${}^{-1}$. PCA–LDA is one of the most commonly used ML model in biomedical RS. PCA reduces the dimensionality of data, and LDA constructs a linear decision boundary for classification. SVM is an ML model which can construct non-linear decision boundaries by using, amongst others, a radial basis function kernel. These are both traditional ML models. A CNN is a DL model which has become popular in medical imaging. It also constructs non-linear decision boundaries and its invariance to certain data inputs make it robust to irrelevant features in the data. Figure A2 provides an overview of the custom CNN developed for this application. These models represent increasingly complex modelling techniques that can be applied to the data. Often with small datasets, a simpler model such as PCA–LDA will perform best as it does not overfit the data, which complex models such as CNNs are prone to do. However, given the complexity of RS data, simpler models may not be able to capture sufficient nuances to be useful. It is not clear which modelling technique would best facilitate discriminating between disease classes and, therefore, a secondary aim of this study is to compare the performance of this range of models.

These models all have hyperparameters, which can be understood as decisions a researcher may make to optimise performance. However, it is possible to over-optimise these hyperparameters so that a model performs well on the research data, but does not generalise to new data. To mitigate against this risk, a repeated nested cross-validation (CV) strategy was used. This allows for hyperparameters to be optimised in an inner CV loop, and then tested against held-out data in an outer CV loop (Figure 1). Nested CV has been shown to reduce a model’s estimated accuracy by as much as 20% in oncological applications of RS, giving a more realistic assessment of the model’s generalisability [8]. A single metric needs to be selected to optimise: the log loss was chosen as it is a proper scoring metric which utilises distributional information, compared to typical binarised metrics such as accuracy [17].

Additionally, we use an occlusion study in order to determine those regions of a spectrum which the CNN uses during classification. This technique involves sequentially “blanking” out regions of an input and returning a prediction. This occluded prediction is then compared to the whole spectrum prediction. A reduction in prediction suggests that the occluded spectral region contained an important feature for the CNN. This technique has, for instance, been used to highlight diagnostically significant brain regions from MRI scans of Alzheimer’s patients [18].

Each sample has at least 40 Raman spectra. The interest in this application is the overall sample label rather than that of individual spectra. Therefore, sample labels were determined by taking all the spectrum-level disease classifications and using a simple majority voting consensus to return an overall label for the sample. Thus, all spectra are used for model construction but a single prediction per sample was obtained. During voting, any ties were to be broken by classifying to the clinically worst disease label, but there were no ties. Data were split into training/test sets on the basis of samples rather than spectra, ensuring that spectra from the same sample were not present in both training and test (or validation) sets, thus maintaining the independence assumption required for model validation. This has been shown to return far more realistic measures of ML performance [8].

Data augmentation was used to supplement the training of the CNN. This is a technique which creates new spectra by replicating existing spectra and adding noise. This helps train the CNN to ignore those perturbed features, reducing over-fitting. This was achieved by adding Poisson noise to the Raman intensity (before normalisation) and adding a random wavenumber shift of no more than +/−6 cm${}^{-1}$ (details in Appendix C). Data augmentation was performed inside the nested CV loops after the data had been split, increasing the training set size by a factor of 8. This technique is a strictly computational method, and does not seek to simulate biomedical Raman spectra.

Analyses were conducted using Python version 3.10 using the Scikit-learn library [19] for the PCA–LDA and SVM models and PyTorch to develop the CNN [20].

Our results are compared to existing diagnostic and screening tests and benchmarks. Current UK guidelines recommend that all CRC patients are offered IHC testing for MMR proteins (a histology-based technique) or MSI-H testing (a genetic-based technique) [21]. If either of these is positive, then subsequent tests are conducted, including genetic testing of germline DNA to detect LS. These tests are two class models, distinguishing in the first instance between dMMR/MSI-H in samples already diagnosed as CRC. To fit into this existing clinical pipeline, two class models were developed using the nested CV strategy described above. The principle performance metrics used to assess IHC and MSI-H testing in the UK guidelines are sensitivity and specificity and these are reported below, alongside the area under the receiver operating characteristic (AUROC) curve.

## 3. Results

The three-fold CV strategy, repeated five times, returned 15 estimates of the performance of each model, allowing for an average performance to be calculated. It was not possible to construct confidence intervals from these results as each CV fold contains over-lapping data, violating the independence assumption required for confidence intervals. In lieu of confidence intervals, standard deviations are reported.

### 3.1. Spectral Data Analysis

Spectra belonging to the same disease class were averaged and are shown in Figure 2. Visual inspection reveals little appreciable difference, as the general composition of the tissues is similar. Together with the standard deviations of the average spectra, this shows how subtle the biochemical differences are between disease classes. To better contrast these subtleties, a difference spectrum was obtained by subtracting the average MSS spectrum from the average MSI-H spectrum (Figure 3). From this, it is possible to infer some biochemical differences between the classes. In particular, peaks at 714, 1081, 1302 and 1445 cm${}^{-1}$ have tentatively been assigned to lipids. Other peak assignments include 1672 cm${}^{-1}$ (cholesterol), 494 cm${}^{-1}$ (glycogen, nucleic acids), 529 cm${}^{-1}$ (amino acids), 732 cm${}^{-1}$ (phosphatidylserine, adenine), 787 cm${}^{-1}$ (nucleic acids), 852 cm${}^{-1}$ (ring-breathing mode of proline, hydroxyproline, tyrosine), 1003 and 1034 cm${}^{-1}$ (phenylalanine, polysaccharides), 1110 cm${}^{-1}$ (lipids, proteins), 1366 cm${}^{-1}$ (tryptophan, lipids, guanine) and 1583 cm${}^{-1}$ (C-C bending mode of phenylalanine). Overall, these indicate differences in nucleic acids, proteins and lipids between the two classes. Band assignments were made using findings contained within the work of Movasaghi et al. [22].

### 3.2. Two-Class Model

Current UK guidelines recommend that all CRC patients are offered IHC testing for MMR proteins or MSI-H testing [21]. If either of these is positive, then subsequent tests are conducted, including genetic testing of germline DNA to detect LS. These tests are two-class models, distinguishing between dMMR/MSI-H in samples already diagnosed as CRC. In this section, a two-class model was developed to this end, consisting of samples that have already been diagnosed as CRC but requiring discrimination between MSI-H and MSS. The principle performance metrics used to assess IHC and MSI-H testing in the UK guidelines are sensitivity and specificity, and they are reported below, alongside the area under the receiver operating characteristic (AUROC) curve. Results from all three ML models are shown in Table 1.

The SVM model performs best in terms of sensitivity at 85.6%, but trades heavily for this with a low specificity of 32.8%, while the PCA–LDA is best for specificity at 62.8%. The CNN performance is between these two traditional ML models. The large standard deviations are likely an artefact of the small sample size. This makes it difficult to draw any conclusions regarding the superiority of any of the models. Despite the CNN being the largest model, and therefore more prone to over-fitting, it returns the lowest variance, indicating a more stable model.

Binary ML models typically do not give a classification but a prediction, a numerical output between 0 and 1, where in this case 0 represents MSI-H and 1 MSS. To calculate sensitivities and specificities, a threshold needs to be applied to the model outputs. A separate, and much larger, study would be required to calibrate the optimal balance of sensitivity and specificity desirable for this application. In lieu of such calibration, the sensitivities and specificities reported use the standard threshold of 0.5. An ROC curve summarises performance over a range of thresholds, giving an indication of how the models may perform under different calibration conditions. The CNN achieves the best AUROC at 0.75. This is often considered a good performance, though this is context-dependent (Figure 4).

### 3.3. Occlusion Study

In Section 3.1, we explored the mean and difference spectra in an attempt to find the biomolecular markers which distinguish LS from AC. However, it is not necessarily the case that a model learns to use those particular features to make its prediction. To elucidate this information, we use an occlusion study. This technique involves sequentially “blanking” out regions of an input and returning a prediction. This occluded prediction is then compared to the whole spectrum prediction. A reduction in prediction suggests that the occluded spectral region contained an important feature for the CNN. This technique has, for instance, been used to highlight diagnostically significant brain regions from MRI scans of Alzheimer’s patients [18].

The region between 680 and 1020 cm${}^{-1}$ seems to be diagnostically significant to the CNN (Figure 5). In particular, the regions between 680 and 710 cm${}^{-1}$ (associated with the ring breathing modes of DNA, C-S bond of methionine and C-N bond of phospholipids and adenine), 800–830 cm${}^{-1}$ (uracil-based ring breathing mode, O-P-O stretching and C${}_{5}$-O-P-O-C${}_{3}$ phosphodiester bands of RNA, PO${}_{2}^{-}$ stretch of nucleic acids as well as C-C stretching in collagen and proline and hydroxyproline) and 870–900 cm${}^{-1}$ (C-C stretching of collagen and C-O-C skeletal mode of monosaccharides, disaccharides and adenine) [22]. Overall, these regions are largely associated with nucleic acids, consistent with findings according to which MSI-H cancers tend to be diploid rather than aneuploid, as well as collagen. The latter is consistent with recent findings according to which *COL11A1* mutations, affecting non-fibrillar collagen expression in the extracellular matrix of MSS colonic and ovarian tissues, may be a useful oncological biomarker [23].

### 3.4. Three-Class Model

Although current practice is to test for dMMR/MSI-H only on confirmed cases of CRC, RS allows for the possibility of screening all suspected CRC samples by using a three-class model at first inspection. If the performance of a model is sufficiently discriminatory between non-cancerous tissue, MSS AC and MSI-H AC, this would bypass the need for the two-step approach currently in practice, whereby samples are first tested for CRC and if positive then undergo a further test to discriminate between MSI-H and MSS. Such a streamlined process may help ameliorate the afore-mentioned lack of surveillance in this regard [5]. To assess this possibility, a three-class model was trained using the same CV strategy outlined in Section 2.3.

Sensitivity, specificity and ROC curves are only defined for binary classification, often implemented as one-vs.-all in multiclass tasks. However, the log loss is reported here (Table 2), to remain consistent with prior results, alongside the more intuitive accuracy and confusion matrices (Figure 6).

The three models return similar accuracies, though the corresponding confusion matrices show that they achieve this by different means; while all models are able to well separate healthy tissues from diseased, the sub-division of the AC cases is more difficult. The SVM does well in correctly identifying MMS with only 4% error, but misclassifies 22% of MSI-H samples as healthy. PCA–LDA and the CNN perform less well discriminating between MSI-H and MSS with 24% error, but only misclassify AC tissue as healthy 4% and 6% of the time, respectively.

The log loss measures the performance of predictions rather than classifications: it compares the probability of belonging to a disease class, rather than a definitive disease label. The lower the log loss, the closer a prediction is to the true disease class. This allows for a more subtle interpretation of performance. The lower log loss of the CNN indicates that this model gives more conservative predictions compared to the other models, particularly the SVM, meaning it is less likely to confidently give incorrect classifications.

## 4. Discussion

A direct comparison with existing testing (i.e., IHC or MSI testing) is difficult as various studies have used different thresholds and definitions for these tests and used different methods for the gold standard genetic testing. The resulting study’s heterogeneity means no meta-analysis is available to provide a statistically pooled estimate of performance [21]. For MSI testing, sensitivities range from 100% to 66.7% [24,25,26] with specificities from 92.5% to 61.1% [24,26]. For IHC testing, sensitivities range from 100% to 80.8% [25,27], and specificities range from 91.9% to 80.5% [27,28].

False positives from both methods have been related to the tissue-processing method. Formalin fixation is known to fragment DNA, causes issues with epitope access due to excessive cross-linking and alters native proteins. These biochemical alterations have also been shown to alter Raman spectra, similarly afflicting the technique’s diagnostic potential [29]. However, RS can easily bypass these alterations by taking spectra from fresh tissues or minimally processed samples, though the degree of any improvement in performance would need empirical corroboration.

The results in this study sit in the lower range of the above sensitivities and specificities, indicating some potential to compete with molecular techniques, but quite some improvement would be needed to establish superiority. Other screening methods based on familial history are also used to identify at-risk patients. The Amsterdam II criteria achieve a sensitivity and specificity of 72% and 78% respectively; the Bethesda protocol 94% and 25%, against which these study’s results are competitive.

DL has also been explored for MSI-H prediction performed on H&E stained samples [4,6]. These take CNNs designed for image classification, which tend to be very large models and require more data to train. These results tend to be given in terms of AUROC, which on average range from 0.77 to 0.93 during internal validation and from 0.60 to 0.89 when applied to an external dataset. This indicates that morphology alone can distinguish MSI-H, a hitherto unused biomarker, though the generalisability of the DL models to external datasets needs improving before clinical adoption. The results of this study show the potential of RS to discriminate between MSS and MSI-H tumours based on molecular information alone. RS allows for the possibility of combining biomolecular and morphological information. While typical image classification takes images as 3D inputs, with red, green and blue channels, a Raman hyperspectral image has several hundreds of channels, which contain biomolecular information.

This study is too small for clinical utility. However, it does demonstrate the predictive capability of RS to this particular clinical problem and therefore motivates the development of larger studies. This study has also confirmed the suitability of CNNs to modelling such data, which has at the least displayed non-inferiority to simpler modelling techniques traditionally used in RS. This would likely improve with larger sample sizes, as DL requires many examples in order to achieve its best possible performance. Consistent with this, AUROC scores assessing the performance of DL to predict MSI-H from H&E slides have been shown to be positively correlated with sample size [6]. Indeed, a related technique to RS, infrared (IR) spectroscopy, has been used to predict MSI-H and achieved a sensitivity and specificity of 100% and 93%, respectively, with a sample size of 100 patients [30].

Another limitation of this study is that the normal, healthy samples were taken from the same patients as the MSS/MMR proficient samples, due to the ethical constraints of obtaining samples from truly healthy colon biopsies. This means that these two classes are not truly independent and there is a risk that the three-class model is consequently biased. Additionally, the three-class model suffers from the fact that the normal samples have been taken from ostensibly healthy regions from patients with confirmed disease. These may well harbour sub-clinical oncogenic mutations that have yet to manifest morphologically, but that may have biochemical antecedents detectable by RS. Hence, the extent to which these samples represent truly healthy tissue is debatable. This problem is common to many oncology RS studies [8]. It has been argued that this approach, called “paired sampling”, reduces interference by individual differences [31], as seen with traditional statistical hypothesis testing. However, it is not clear that ML models similarly benefit from this effect. Conversely, sampling from the same patients likely hinders the generalisability of models trained in this manner when deployed on truly independent samples. The two-class model is unaffected by this potential bias as it did not use the healthy samples. Another limitation is that this study only took resected tissue and no biopsied tissue. It has been shown that models trained to detect MSI-H with H&E samples on one type suffer when trying to predict tissue taken by the other method [32]. Finally, consensus agreement between pathologists is the gold standard for labelling histology slides for ML, but this study only obtained single pathologist labelling.

## 5. Conclusions

This study is the first of its kind to carry out a preliminary investigation into the use of RS as a clinical diagnostic tool in discriminating MSS AC from MSI-H AC in FFPE colonic tissues. From a very small sample size (10 samples per group), promising results were achieved with the use of ML models, which show that a reasonable degree of discrimination is possible from samples that appear to be spectrally very similar. This is the first proof-of-principle study of its kind that is both label-free and has a rapid sample turnaround time.

Through post hoc analysis of the DL model, diagnostically relevant molecular biomarkers have been implicated, which may distinguish MSS from MSI-H, with nucleic acids and collagen being particularly pertinent. The DL model was able to achieve equivalent performance with screening methods based on familial history, despite the low sample size, though this is not yet competitive with molecular testing.

## Figures and Tables

**Figure 1 cancers-15-01720-f001:**
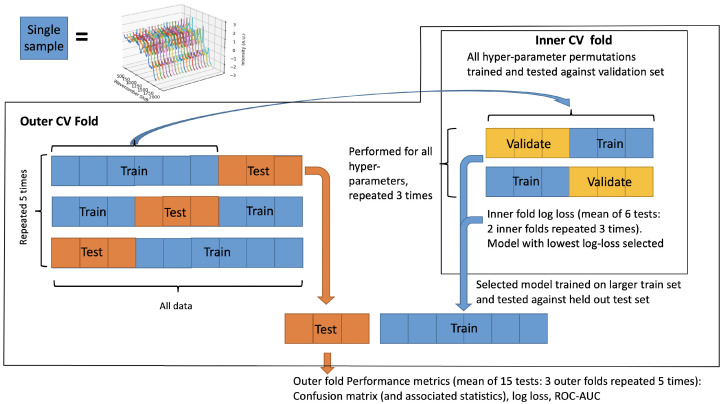
Nested CV strategy.

**Figure 2 cancers-15-01720-f002:**
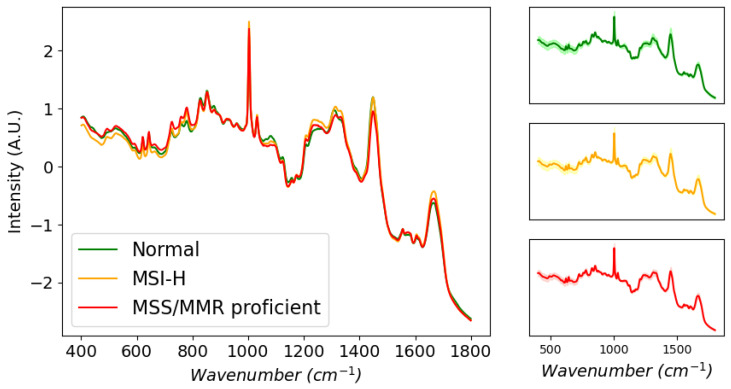
Average normalised spectrum by class. Right panels, average spectra with shaded areas indicating 1 standard deviation.

**Figure 3 cancers-15-01720-f003:**
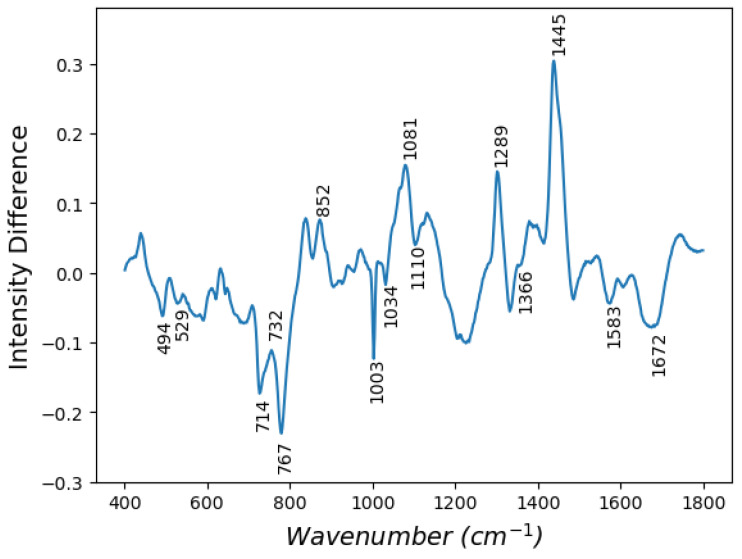
Difference spectrum: MSI-H minus MSS. Numbers indicate peaks mentioned in the text.

**Figure 4 cancers-15-01720-f004:**
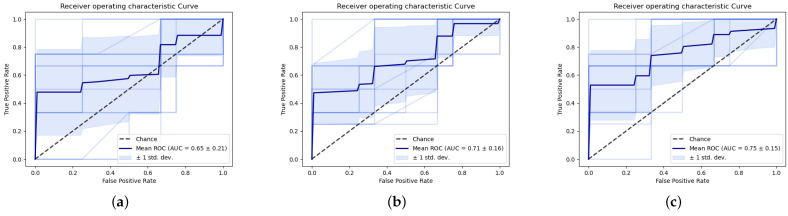
Receiver Operating characteristic curve for (**a**) PCA–LDA (**b**), SVM (**c**), CNN. Bold lines indicate mean ROC, pale lines performance for individual folds and shaded area 1 standard deviation.

**Figure 5 cancers-15-01720-f005:**
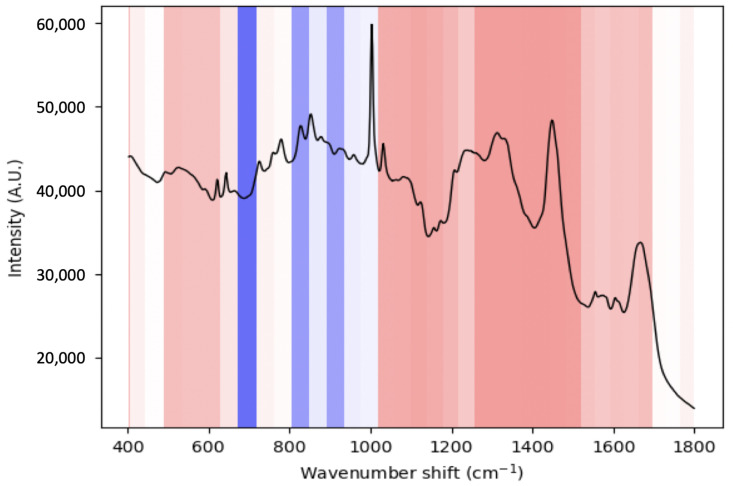
Occlusion study: Blue indicates drops in performance due to occlusion. The stronger the shade, the larger the drop in performance.

**Figure 6 cancers-15-01720-f006:**
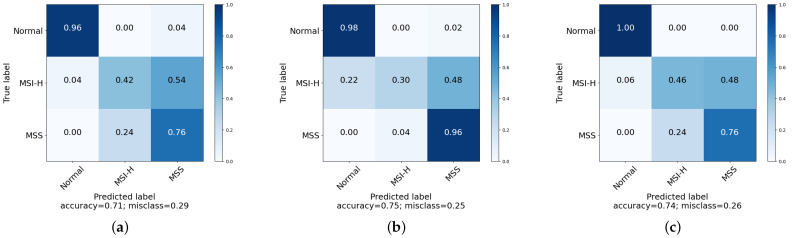
Confusion matrix for (**a**) PCA–LDA (**b**), SVM (**c**), CNN.

**Table 1 cancers-15-01720-t001:** Two-class models: mean sensitivity, specificity and AUROC across all folds +/−1 standard deviation.

	PCA–LDA	SVM	CNN
**Sensitivity**	70.0% +/− 36.1	85.6% +/− 21.0	73.0% +/− 10.0
**Specificity**	62.8% +/− 27.5	32.8% +/− 15.7	48.9% +/− 12.5
**AUROC**	0.65 +/− 0.21	0.71 +/− 0.16	0.75 +/− 0.15

**Table 2 cancers-15-01720-t002:** Three class models: mean log loss and accuracy across all folds +/−1 standard deviation.

	PCA–LDA	SVM	CNN
**Log Loss**	0.66 +/− 0.17	0.80 +/− 0.07	0.54 +/− 0.17
**Accuracy**	71.3% +/− 8.8	74.7% +/− 7.2	74.0% +/− 12.0

## Data Availability

The data and models presented in this study are available upon request from the corresponding author (G.M.H.T.).

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
