# Peer review of "Deep Learning Applied to Raman Spectroscopy for the Detection of Microsatellite Instability/MMR Deficient Colorectal Cancer"

_cancers, 2023, doi:10.3390/cancers15061720_

Round 1

Reviewer 1 Report

1. You should not use subsections in the Introduction.

2. Within one tissue sample, 50 spectra were taken, then averaging was carried out, did I understand correctly? Was tissue homogeneity assessed before? Or were there differences?

3. A total of 10 samples in each group, this is a small sample to obtain reliable conclusions.

4. What spectral parameters are used for processing by multidimensional methods: intensity of absorption bands, presence/absence of a band, etc.?

5. With what accuracy can a histologist distinguish between these types of specimens? It is necessary to understand with what to compare the results of this work.

Author Response

Thanks for the feedback to help improve our manuscript. We have made the following changes based on your recommendations, addressed point by point below.

We hope that we have sufficiently understood and addressed your comments. Attached is a PDF of the amended manuscript with changes from both reviewers highlighted so that you may see changes in situ.

1.) You should not use subsections in the Introduction.

Subsections removed from the Introduction.

2.) Within one tissue sample, 50 spectra were taken, then averaging was carried out, did I understand correctly? Was tissue homogeneity assessed before? Or were there differences?

No averaging was done, but rather all spectra in any single sample were classified per spectrum, then the most common disease class from all spectra in the sample was chosen as the overall label. Thus sample heterogeneity has been incorporated into the model. The following text was added to the text to clarify: 

Therefore, sample labels were determined by taking all the spectrum level disease classifications and using a simple majority voting consensus to return an overall label for the sample. Thus, all spectra are used for model construction but a single classification per sample is obtained. During voting, any ties were to be broken by classifying to the clinically worst disease label, but there were no ties. (LINE 180-185)

3.) A total of 10 samples in each group, this is a small sample to obtain reliable conclusions.

We agree the sample size is very small. This limitation has been made explicit by highlighting the exploratory nature of this work in the introduction: 

The exploratory nature of this study seeks only to assess whether RS applied to this particular clinical problem merits larger studies, to highlight some of the methodological considerations for such a larger study and to explore potential clinical uses such as screening or diagnosis. (LINE 95-99) 

and the following to the discussion/limitations: 

This study is too small for clinical utility. However, it does demonstrate the predictive capability of RS to this particular clinical problem and so motivates the development of larger studies. (LINE 343-347) 

4.) What spectral parameters are used for processing by multidimensional methods: intensity of absorption bands, presence/absence of a band, etc.?

The Raman effect is a scattering phenomenon, thus the bands represent scattered photons of shifted wavelengths. The following text was added to the text to clarify: 

A Raman spectrum represents the change in photon wavenumber from a monochromatic light source along the x-axis and the intensity (i.e. number of photons) thus scattered on the y-axis. (LINE 66-68)

And

using SNV normalised Raman spectra truncated to the range 400 -1800. (LINE 146)

5.) With what accuracy can a histologist distinguish between these types of specimens? It is necessary to understand with what to compare the results of this work.

Standard IHC and MSI testing sensitivities and specificities for MSI have been reported. They have also been compared to screening methods based on family history. In addition, our results are compared to recent DL techniques using H&E slide images. The following text was amended to the text to clarify: 

Our results are compared to existing diagnostic and screening tests and benchmarks. Current UK guidelines recommend that all CRC patients are offered IHC testing for MMR proteins (a histology based technique) or MSI-H testing (a genetic based technique). (LINE 200-202)

Reviewer 2 Report

Dear authors and editor,

Thanks for the opportunity to review this manuscript by Blake et al., entitled "Deep learning applied to Raman Spectroscopy for the detection of microsatellite instability / MMR deficient colorectal cancer." This work explored a convolutional neural network approach to analyze Raman spectroscopy data and detect MMR deficiency in histological samples of colorectal cancer.

We find this manuscript reasonably novel. Although CRC detection using the combination of RS+CNN has been previously published, for instance in
Cao, Z., Pan, X., Yu, H., Hua, S., Wang, D., Chen, D. Z., ... & Wu, J. (2022). "A deep learning approach for detecting colorectal cancer via Raman spectra." BME Frontiers.,
there appears yet to be a report on the specific detection of the MMR anomalies using this combination. The approach taken and results presented seem to be generally sound. This work could be of value to audiences interested in immuno-oncology and deep learning.

We have a few specific question/comments, detailed as follows.

1. Section 3.4: The three-class model comparing healthy tissue, MSS and dMMR, strikes us as somewhat odd. It was not immediately obvious what the appropriate clinical setting this model is applicable to. We felt the manuscript can benefit from some clarification in this regard.

If this is used in a screening setting for at-risk population (e.g. screening colonoscopy), wouldn't it make more sense to use a two class (healthy vs cancer) model, and defer the MSS vs dMMR discrimination to clinical work-up once there's a confirmed cancer diagnosis? If used in a diagnostic setting to analyze CRC histology in patients, would it be necessary to have a "healthy" category?

2. Page 9 line 297: Here it was pointed out that the healthy and diseased histological samples were not truly mutually independent since they were collected from the same patients. We felt another major downside was that the so-called "healthy" samples came from a selected cohort who is predisposed to (and actually diagnosed with) colorectal cancer, and they may harbor certain somatic or germline oncogenic mutations despite histologically normal-appearing; additionally, the methods described that these tissues came from resection margins. Whether these samples can truly represent healthy tissue in the general population is unknown, and this should be discussed.

3. Table 1 & Page 7 line 205: Whereas one can appreciate the difference in performance of each CNN & ML approach by the area under ROC, we felt it wasn't so much the case with sensitivity/specificity.

The reported sensitivity/specificity for each approach, in its current form, makes it appear like three distinct diagnostic tests and therefore rather challenging to judge which one better satisfies outstanding clinical needs. Here a fixed threshold was chosen. Shouldn't the threshold be determined based on both the prevalence of dMMR in CRC all comers and the proposed clinical scenario?

Is the proposed approach intended to be a independent diagnostic tool for dMMR detection, or a filtering tool -- for instance, to select patients for a confirmatory NGS? The requirement to minimize type 1/2 errors, and therefore desirable sensitivity/specificity profile would be different in the former vs latter. Does it make sense to apply a desired sensitivity to all three, and compare their differences in specificity, or vice versa?

4. We felt some text in the Results sections belongs more appropriately to the Methods section. For example, (Page 5 line 170) "Analyses were conducted ...  to develop the CNN"; (Page 6 line 189) "Current UK guidelines recommend ... two class models were developed ..."; (Page 7 line 211) "This technique involves ... Alzheimer’s patients."

5. Page 2 line 67: "... high dimensionality ..." The spectroscopic data presented, e.g. in Figure 2, seem to be 1D. Could it be clarified what it means by "high dimensional"?

6. Page 3 line 121: "Saturated spectra were discarded at the time of acquisition and a new spectrum obtained." Were any imaging parameters modified to eliminate spectral saturation? This should be described.

7. Page 11 line 361: "13 Baseline correction methods were employed ... and EMSC." These baseline correction methods were evaluated but not employed, correct? Also if these methods are listed, it would be good to provide citations as readers are not necessarily familiar with them.

Best regards,

Author Response

Thanks for the detailed feedback to help improve our manuscript. We have made the following changes based on your recommendations, addressed point by point below.

We hope that we have sufficiently understood and addressed your comments. Attached is a PDF of the amended manuscript with changes from both reviewers highlighted so that you may see changes in situ.

1.) Section 3.4: The three-class model comparing healthy tissue, MSS and dMMR, strikes us as somewhat odd. It was not immediately obvious what the appropriate clinical setting this model is applicable to. We felt the manuscript can benefit from some clarification in this regard.

If this is used in a screening setting for at-risk population (e.g. screening colonoscopy), wouldn't it make more sense to use a two class (healthy vs cancer) model, and defer the MSS vs dMMR discrimination to clinical work-up once there's a confirmed cancer diagnosis? If used in a diagnostic setting to analyze CRC histology in patients, would it be necessary to have a "healthy" category?

The two class model is indeed just discriminating between MSS and dMMR as suggested and as is standard process at the moment. To clarify this the text has been amended to:

These tests are two class models, distinguishing between dMMR/MSI-H in samples already diagnosed as CRC. In this section a two class model was developed to this end, consisting of samples that have already been diagnosed as CRC but requiring discrimination between MSI-H and MSS. (LINE 237-241)

The three class model was an exploration of whether a single model would be sufficient to discriminate MSS from dMMR even at screening, thus bypassing the need for further work-up, which may help elevate the problem identified in the literature that testing for MSI-H and MSS is poor. To make this more clear, the text has been amended to:

Although current practice is to test for dMMR/MSI-H only on confirmed cases of CRC, RS allows for the possibility of screening all suspected CRC samples by using a three class model at screening. If the performance of a model is sufficiently discriminatory between non-cancerous tissue, MSS AC and MSI-H AC, this would bypass the need for two-step approach currently in practice whereby samples are first tested for CRC and if positive then under going a further test to discriminate between MSI-H and MSS pathways. Such a streamlined process may help ameliorate the afore mentioned lack of surveillance in this regard.  (LINE 284-291)

2.) Page 9 line 297: Here it was pointed out that the healthy and diseased histological samples were not truly mutually independent since they were collected from the same patients. We felt another major downside was that the so-called "healthy" samples came from a selected cohort who is predisposed to (and actually diagnosed with) colorectal cancer, and they may harbor certain somatic or germline oncogenic mutations despite histologically normal-appearing; additionally, the methods described that these tissues came from resection margins. Whether these samples can truly represent healthy tissue in the general population is unknown, and this should be discussed.

We agree with your analysis, though we feel it is mitigated as any underlying genetic changes in MSS patients are not MMR related, which is the primary focus of this research. However, we have amended the text to address this limitation in more detail to...

Additionally, the three class model suffers from the fact that the normal samples have been taken from ostensibly healthy regions from patients with confirmed disease. These may well harbour sub-clinical oncogenic mutations that have yet to manifest morphologically, but that may have biochemical antecedents detectable by RS. Hence, the extent to which these samples represent truly healthy tissue is debatable. This problem is common to many oncology RS studies \cite{diagnostics12061491}. It had been argued that this approach, called 'paired sampling' reduces interference by individual differences \cite{ma2021classifying}, as seen with traditional statistical hypothesis testing. However, it is not clear that ML similarly benefits from this effect. Conversely, sampling from the same patients likely hinders the generalisability of models trained in this manner when deployed on truly independent samples. (LINE 357-366)

3.) Table 1 & Page 7 line 205: Whereas one can appreciate the difference in performance of each CNN & ML approach by the area under ROC, we felt it wasn't so much the case with sensitivity/specificity.

The reported sensitivity/specificity for each approach, in its current form, makes it appear like three distinct diagnostic tests and therefore rather challenging to judge which one better satisfies outstanding clinical needs. Here a fixed threshold was chosen. Shouldn't the threshold be determined based on both the prevalence of dMMR in CRC all comers and the proposed clinical scenario?

Is the proposed approach intended to be a independent diagnostic tool for dMMR detection, or a filtering tool -- for instance, to select patients for a confirmatory NGS? The requirement to minimize type 1/2 errors, and therefore desirable sensitivity/specificity profile would be different in the former vs latter. Does it make sense to apply a desired sensitivity to all three, and compare their differences in specificity, or vice versa?

Sensitivity and specificity metrics were included because existing guidelines use them, facilitating comparisons to existing diagnostic tests. We agree that interpreting the results of these metrics is not straightforward. The text was amended to make this more clear: 

This makes it difficult to draw any conclusions regarding the superiority of any of the models. (LINE 248-249)

The paper explored three distinct ways of modelling the same diagnostic test (i.e. RS), hence the reported metrics are comparisons between ML models as opposed to different diagnostic tests, to help determine which of these models is best if using RS as a diagnostic test. This was made more explicit in the Materials and Methods section: 

These models represent increasingly complex modelling techniques that can be applied to the data. Often with small datasets a simpler model like PCA-LDA will perform best as it does not overfit the data, which complex models like CNNs are prone to do. However, given the complexity of RS data, simpler models may not be able to capture sufficient nuances to be useful. It is not clear which modelling technique would best facilitate discriminating between disease classes and so a secondary aim of this study is to compare the performance of this range of models. (LINE 153-160)

And in the discussion section: 

This study has also confirmed the suitability of CNNs to modelling such data, which has at the least displayed non-inferiority to simpler modelling techniques traditionally used in RS. (LINE 343-347)

The threshold applied is to the ML models rather than the RS data itself. This is something done implicitly in all ML models whether or not it is stated in a paper, but we state it here explicitly for the sake of transparency. We can omit this is you feel it is an unnecessary detail as we use the same value as everyone else, but we have also amended the text to make this clearer: 

Binary ML models typically do not give a classification but a prediction, a numerical output between 0 and 1, where in this case 0 represents MSI-H and 1 MSS. To calculate sensitivities and specificities a threshold needs to be applied to the model outputs. A separate, and much larger, study would be required to calibrate the optimal balance of sensitivity and specificity desirable for this application. In lieu of such calibration the sensitivities and specificities reported use the standard threshold of 0.5. (LINE 252-257)

4.) We felt some text in the Results sections belongs more appropriately to the Methods section. For example, (Page 5 line 170) "Analyses were conducted ...  to develop the CNN"; (Page 6 line 189) "Current UK guidelines recommend ... two class models were developed ..."; (Page 7 line 211) "This technique involves ... Alzheimer’s patients."

These have all been moved to the appropriate locations in the methods section: 

Analyses were conducted… has been moved to LINE 198. 

Current UK guidelines recommend… to LINE 201. 

This technique involves… has been moved to LINE 173.

  1. Page 2 line 67: "... high dimensionality ..." The spectroscopic data presented, e.g. in Figure 2, seem to be 1D. Could it be clarified what it means by "high dimensional"?

Now it’s been pointed out, we can see how this is confusing. To clarify this the text has been amended to: 

Additionally, the inherent complexity and high dimensionality (number of discrete wavenumber points per spectrum) of biomedical RS data (LINE 77-78)

6.) Page 3 line 121: "Saturated spectra were discarded at the time of acquisition and a new spectrum obtained." Were any imaging parameters modified to eliminate spectral saturation? This should be described.

No imaging parameters were used for saturation detection, but visual inspection of the spectra at the time of acquisition was performed, and saturated spectra removed if present and the measurement repeated. To clarify this the text has been amended to: 

Spectra were visually inspected at the time of acquisition and any saturated spectra were discarded and a new spectrum obtained from a different region. (LINE 137-138).

7.) Page 11 line 361: "13 Baseline correction methods were employed ... and EMSC." These baseline correction methods were evaluated but not employed, correct? Also if these methods are listed, it would be good to provide citations as readers are not necessarily familiar with them.

Correct. Appendix A was a sub-study within itself, assessing the need for baseline correction. Four baseline methods, with varying parameters, were used. We can remove this appendix if you feel it distracts: it was included to justify the choice of not using baseline correction in the actual analysis. Citations for all 4 baseline methods used are given in the appendix, but these have also been added to the main text. (LINE 141). 

Additionally the appendix text has been altered to be more clear to: 

13 Baseline correction methods were evaluated. (LINE 431)

And:

We took these four techniques, two of which extend over a range of polynomial orders, and applied them to the dataset. (LINE 424-425) 

Round 2

Reviewer 1 Report

The authors responded to the reviewer's comments and made corrections to the manuscript. I think that in its present form the article can be recommended for publication.